# Involvement of Autonomic Nervous System in New-Onset Atrial Fibrillation during Acute Myocardial Infarction

**DOI:** 10.3390/jcm9051481

**Published:** 2020-05-14

**Authors:** Audrey Sagnard, Charles Guenancia, Basile Mouhat, Maud Maza, Marie Fichot, Daniel Moreau, Fabien Garnier, Luc Lorgis, Yves Cottin, Marianne Zeller

**Affiliations:** 1Cardiology Department, University Hospital, 21000 Dijon, France; audrey.sagnard@chu-dijon.fr (A.S.); basile.mouhat@gmail.com (B.M.); maud.maza@chu-dijon.fr (M.M.); marie.fichot@chu-dijon.fr (M.F.); daniel.moreau@u-bourgogne.fr (D.M.); fabien.garnier@chu-dijon.fr (F.G.); luc.lorgis@chu-dijon.fr (L.L.); yves.cottin@chu-dijon.fr (Y.C.); marianne.zeller@u-bourgogne.fr (M.Z.); 2PEC 2 EA 7460, University of Burgundy and Franche-Comté, 21000 Dijon, France

**Keywords:** acute myocardial infarction, atrial fibrillation, heart rate variability, autonomic nervous system

## Abstract

**Background:** Atrial fibrillation (AF) is common after acute myocardial infarction (AMI) and associated with in-hospital and long-term mortality. However, the pathophysiology of AF in AMI is poorly understood. Heart rate variability (HRV), measured by Holter-ECG, reflects cardiovascular response to the autonomic nervous system and altered (reduced or enhanced) HRV may have a major role in the onset of AF in AMI patients. Objective: We investigated the relationship between autonomic dysregulation and new-onset AF during AMI. Methods: As part of the RICO survey, all consecutive patients hospitalized for AMI at Dijon (France) university hospital between June 2001 and November 2014 were analyzed by Holter-ECG <24 h following admission. HRV was measured using temporal and spectral analysis. **Results:** Among the 2040 included patients, 168 (8.2%) developed AF during AMI. Compared to the sinus-rhythm (SR) group, AF patients were older, had more frequent hypertension and lower left ventricular ejection fraction LVEF. On the Holter parameters, AF patients had higher pNN50 values (11% vs. 4%, p < 0.001) and median LH/HF ratio, a reflection of sympathovagal balance, was significantly lower in the AF group (0.88 vs 2.75 p < 0.001). The optimal LF/HF cut-off for AF prediction was 1.735. In multivariate analyses, low LF/HF <1.735 (OR(95%CI) = 3.377 (2.047–5.572)) was strongly associated with AF, ahead of age (OR(95%CI) = 1.04(1.01–1.06)), mean sinus-rhythm rate (OR(95%CI) = 1.03(1.02–1.05)) and log NT-proBNP (OR(95%CI) = 1.38(1.01–1.90). **Conclusion:** Our study strongly suggests that new-onset AF in AMI mainly occurs in a dysregulated autonomic nervous system, as suggested by low LF/HF, and higher PNN50 and RMSSD values.

## 1. Introduction

Atrial fibrillation (AF) is one of the most common cardiovascular (CV) diseases worldwide, with an increasing global burden associated with the ageing of the population. AF frequently occurs in patients with acute myocardial infarction (AMI), with an incidence ranging from 6% to 21% [1], and is associated with an increased risk of death and rehospitalisation for heart failure, which in turn have major economic consequences [2]. AF leads to atrial remodeling through an alteration of the electrical and structural properties of the atria, thus facilitating the maintenance and recurrence of AF [3]. Several risk factors for AF have already been identified in various populations, but the predictive power of individual risk factors is still far from accurate. Heart rate variability (HRV), measured by Holter-ECG, reflects cardiovascular response to the autonomic nervous system (ANS) and altered (reduced or enhanced) HRV may have a major role in the onset of AF in AMI patients with coronary artery disease (CAD) [4]. However, in the setting of AMI, the role of the ANS is uncertain and data are scarce [5]. The aim of this retrospective analysis of a large database of patients was to evaluate whether autonomic dysfunction (measured by HRV) could be associated with new-onset AF in AMI: is it an epiphenomenon related to sympathetic activation linked to the severity of the infarction, or is it related to parasympathetic dependence which would suggest chronic electrical and anatomical atrial remodeling and therefore risk of recurrence?

## 2. Experimental Section

### 2.1. Patients

The participants were recruited from the RICO (obseRvatoire des Infarctus de Côte-d’Or) database, a regional registry for cases of acute myocardial infarction (MI). Briefly, RICO collects data from all patients hospitalized for acute MI in all public or privately funded hospitals of one department in the east of France [6]. The present study included all consecutive patients admitted between 20th June 2001 and 2nd November 2014 (>18 years old) who underwent a 24-h Holter ECG recording during their coronary care unit stay. Patients with a history of AF (n = 94) were excluded, we checked for each patient admission his/her previous medical history, baseline treatments and medical recording in the hospital files. In case of suspicion of previous AF that cannot be confirmed by the patient (anticoagulant therapy, antiarrhythmic drug) we systematically called the general practitioner or the cardiologist of the patient to explain this drug prescriptions. A flow chart reporting the inclusion and exclusion criteria is shown in Figure 1. The present study complied with the Declaration of Helsinki and was approved by the ethics committee of the University Hospital of Dijon. Each patient provided written consent before participation.

### 2.2. Data Collection

Patient data were collected from the RICO database: continuous electrocardiographic monitoring (CEM) data (rhythm status during AMI), cardiovascular risk factors, clinical data, type of AMI, acute management, acute and discharge medications, biological data, echocardiography data including LVEF (with a cut-off at 40% using Simpson’s method for more clinical relevance) and left atrial dimensions as previously described [7].

### 2.3. AF Definition

AF was diagnosed in accordance with the current European Society of Cardiology Guidelines as absolutely irregular RR intervals and no discernible, distinct P waves and an episode duration of at least 30 s. Flutter episodes were included as AF episodes [8].

### 2.4. Biological Data

Blood samples were drawn on admission. Plasma creatinine levels were measured on a Vitros 950 analyzer (Ortho Clinical Diagnostics, Rochester, New York, USA). Glomerular filtration rate was calculated with the Chronic Kidney Disease Epidemiology Collaboration (CKD-EPI) formula. C-reactive protein (CRP) was determined on a dimension Xpand (Dade Behring, Deerfield, Illinois, USA) using enzymatic methods. CRP level was dichotomized into high and low categories at 3 mg/L for more clinical relevance.

### 2.5. Holter ECG Data

If the patient consented to undergo a 24-h (Holter) ECG, it was done in the coronary care unit within the 24 h following admission. Holter monitoring was based on device availability [6], patient consent and expected hospital stay >48 h within our cardiology department to be able to obtain Holter results before hospital discharge. Long ECG tracing was recorded and analyzed by two experienced observers using a Syneflash digital recorder Holter (Ela medical and Spieder Viers, le Plessis Robinson, France), with seven surface electrode signals (acquisition sampling rate: 1000 Hz). After classifying the QRS morphology, the RR intervals (longest and shortest) were confirmed manually until no QRS sequences were incorrectly labelled. Only sequences with normal QRS characteristics during 24 h (sinus rhythm) were analysed for HRV study. HRV was addressed from the time or frequency domain in accordance with the 1996 guidelines of the ESC Task Force [9]: Time domain variables: (1) rMSSD: root mean square of successive differences in NN intervals is considered an estimate of the short-term components of HRV, which correspond to parasympathetic activity. (2) pNN50: proportion derived by dividing NN50 (the number of interval differences of successive NN intervals greater than 50 ms) by the total number of NN intervals. This is a measure of parasympathetic activity. (3) SDNN: the standard deviation of all intervals between adjacent QRS complexes resulting from sinus node depolarization (NN), i.e., the square root of variance, reflects the cyclic components responsible for variability in the period of recording and is considered as an estimate of overall HRV, encompassing vagal and sympathetic influences [10,11].

Frequency domain variables: Fast Fourier transform was used to convert the different successive RR intervals in the frequency domain. Low frequencies (LF), between 0.04 and 0.15 Hz, are affected by both vagal and sympathetic activity, whereas high frequencies (HF), between 0.15 and 0.4 Hz, are affected by vagal tone. The LF/HF ratio is therefore considered an indicator of sympathovagal balance; oscillations in very low frequencies VLF (range 0.00 to 0.04 Hz) reflect peripheral vasomotor regulation. Total power (TP), combining the sum of all the frequencies, is a global measure of ANS activity [10,11].

### 2.6. Statistical Analyses

Results were expressed as medians [IQR] for continuous variables and as percentages for dichotomized variables. Normality was tested with the Kolmogorov-Smirnov test. For continuous data, we used the Mann-Whitney test. Categorical variables were compared with the χ^2^ or Fisher exact test. A p value of less than 0.05 was considered statistically significant. Since the NT-proBNP values do not follow a normal distribution, a logarithmic transformation was performed. To assess discrimination for AF recurrences, we examined the area under the receiver-operating characteristic (ROC) curve (plot of sensitivity versus 1—specificity for all possible cut-off values to classify predictions) for the LF-HF with the best sensitivity, specificity, positive predictive value and negative predictive value according to the Youden index. Multivariate logistic regression models were built to estimate the odds ratio (OR) of in-hospital AF and of a LF/HF ratio cut-off value. Variables that met the statistical significance threshold of 5% in univariable analysis were included in the multivariate models. LVEF was not included in the final model because of collinearity with NT-proBNP. To improve the robustness of results, the AF patients were matched 1:1 with the SR patients using nearest neighbor matching on the linear propensity score with a tolerance of 0.02. The same logistic regression model was used to compare AF with SR patients. The statistical tests were performed with SPSS software version 26 (IBM Corp., Armonk, NY, USA).

## 3. Results

### 3.1. Patient Characteristics

Of the 14,270 patients in the RICO database, 2040 (14.3%) met the main inclusion criteria of having a 24-h Holter ECG recording and no prior AF (Figure 1).

AF was identified in 168 (8.2%) of the included patients. Table 1, Table 2, Table 3 and Table 4 summarize the characteristics of the study population. 

Patients from the AF group were almost 10 y older (77 vs. 64 y; *p* < 0.001), more frequently hypertensive (68% vs 49%; *p* < 0.001) and less likely to smoke (15% vs. 35%; *p* < 0.001) then the rest of the study population. The pNN50 values of the AF group were almost thrice higher (11% vs. 4%; *p* < 0.001), their rMSSD values were higher (45 vs. 27 ms; *p* < 0.001) and the HR by Holter ECG was faster (73 vs. 66 beats/min; *p* < 0.001). More AF patients had a LF/HF ratio < 1.735% (75% vs. 30%; *p* < 0.001). High creatinine (98 vs. 87 μmol/L; *p* < 0.001), glycaemia (7.92 vs. 7.00 mmol/L; *p* < 0.001) and NT-proBNP levels (2450 vs. 542 pg/mL; *p* < 0.001) were observed in AF patients. They were also more likely to have a history of cardiovascular disease, including CAD, stroke, and renal failure. Accordingly, they were more likely to be taking chronic CV medications such as beta blockers and amiodarone (medication used for a history of ventricular arrhythmia (no atrial fibrillation ECG traces in their medical records)). The other admission parameters (including diabetes, time to admission, and troponin Ic peak) were not significantly different except for multivessel disease.

### 3.2. ROC Curve

The optimal cut-offs for continuous test variables were determined from the ROC curve, which was used to estimate the optimal threshold value of LF-HF. The best LF/HF value to characterize our population according to AF occurrence was a LF/HF ratio <1.735, with an AUC of 0.73 (95% CI (0.69–0.78); *p* < 0.001), sensitivity of 69% and specificity of 70% (Figure 2).

### 3.3. LF/HF Determinants: Multivariate Analysis

In multivariate analysis, only age, female sex and diabetes were associated with low LF/HF, therefore excluding the influence of treatments such as beta blockers or the severity of AMI on this ANS parameter.

### 3.4. AF Determinants in Acute Myocardial Infarction

In univariate analysis (Table 5) the risk factors for developing AF in the acute phase of infarction were: female sex, age, hypertension, smoking HR on Holter, CRP > 3 mg/L, eGFR, log-NTproBNP, chronic use of ARB/ACE inhibitors and chronic use of beta-blockers.

In multivariate analysis, the independent risk factors for developing AF were age (OR 1.05 (1.03–1.07); *p* < 0.001), HR (OR 1.04 (1.02–1.05); *p* < 0.001) and log NT-proBNP (OR: 1.48(1.10–1.99, *p* = 0.010)) with a good predictive performance. 

The addition of the LF/HF < 1.735 variable significantly improved our ability to predict in-hospital AF (OR 3.38 (2.05–5.57); *p* < 0.001).

Moreover, after 1:1 propensity score matching (on age, sex, previous hypertension, previous stroke, BMI, LVEF), LF/HF ratio <1.735 (OR 3.49 (2.03–5.99), *p* < 0.001) remained independently associated with the new-onset of AF during AMI.

### 3.5. Echocardiographic Parameters of Left Atrium

We performed a subgroup analysis using left atrial (LA) echocardiographic parameters in patients for whom these parameters were available (n = 121 for LA diameter, 117 for LA area and n = 100 for LA volume). We started by conducting a univariate analysis to identify the LA size variable that could most powerfully predict AF. Next, we added the variable to a bivariate model and observed whether LF/HF remained independently associated with AF after adjustment on left atrial size. In univariate analysis, the only LA size parameter that was a predictor of in-hospital AF was LA volume (OR 1.03 (1.00–1.05); *p* < 0.001). Among the patients included in the subgroup analysis, eight had a new-onset of AF during AMI. However, after bivariate analysis, neither LA volume nor LH/HF remained significantly associated with AF (*p* = 0.062 for both variables). Collinearity between the variables was not significant (variation inflation factor = 1.07).

## 4. Discussion

The results of our large, population-based study indicate that a low LF/HF (<1.735) ratio was strongly associated with new-onset AF during AMI. Indeed, investigation of the median LF/HF ratio revealed that the sinus rhythm group and the AF group had a marked difference in sympathovagal balance. In our population, 75% of AF patients had a LF/HF ratio < 1.735, compared to 30% of patients in the sinus rhythm group. Lower values of LF/HF ratio are thought to reflect decreased sympathetic activity and/or increased parasympathetic activity, i.e., above all, an imbalance of this sympathovagal tone.

### 4.1. AF in Acute Myocardial Infarction

It is still not known whether AF in AMI is promoted by acute activation of the sympathetic nervous system and is therefore a reversible arrhythmia, or if it occurs on a pre-existing atrial substrate that is prone to chronic dysregulation of ANS, indicating potential recurrence. This question is of particular interest considering the potential therapeutic consequences, particularly the initiation of oral anticoagulation therapy on top of the dual antiplatelet treatment prescribed to AMI patients. The role of autonomic tone in the genesis of atrial arrhythmia has been clinically recognized for many years, but autonomic modulation is extremely complex to characterize and quantify [4]. In clinical cardiology, the main tool to evaluate ANS activity is the analysis of HRV parameters on continuous ECG recordings [9]. The frequency-domain HRV parameters obtained by spectral analysis are considered the most useful parameters for addressing the sympathetic/parasympathetic balance. The HF components are thought to primarily reflect vagal tone, whereas the more complex LF components probably reflect sympathetic activity [4]. Both the parasympathetic and the sympathetic nervous systems have been shown to play a role in AF. Amar et al., showed that the onset of AF was preceded by a primary increase in the sympathetic drive, followed by marked modulation toward vagal predominance [12]. Our results suggest that there appears to be an ANS dysregulation prior to AMI, resulting in a paradoxical ANS response on the parasympathetic side where we would have expected an over-expression of the sympathetic system only.

### 4.2. LF/HF Ratio Findings

The ANS plays a central pathophysiological role in the initiation and progression of AF [13]. Power spectral analysis of the beat-to-beat variations of heart rate or the heart period (R–R interval) has become widely used to quantify cardiac autonomic regulation. This technique partitions the total variance (the “power”) of a continuous series of beats into its frequency components, typically identifying two main peaks: low frequency (LF), 0.04–0.15 Hz, and high frequency (HF) 0.15–0.4 Hz [9]. The HF peak is widely believed to reflect cardiac parasympathetic nerve activity while the LF, although more complex, is often assumed to have a dominant sympathetic component. Based upon these assumptions, Pagani and colleagues suggested that the ratio of LF to HF (LF/HF) could be used to quantify the changing relationship between sympathetic and parasympathetic nerve activities (i.e., the sympathovagal balance) in both healthy and diseased organisms [14]. The clinical determinants influencing HRV values were evaluated from Framingham’s study, which showed that age and HR were the two main determinants of HRV in healthy subjects, in addition to sex and smoking [15]. The LF/HF ratio has gained wide acceptance as a tool for assessing cardiovascular autonomic regulation where increases in LF/HF are assumed to reflect a shift towards “sympathetic dominance” and decreases indicate “parasympathetic dominance” [9]. Moreover, it has been proved that non-linearity of neural modulation of cycle length may result in an intrinsic rate-dependency of autonomic indexes, with the exception of normalized frequency-domain indexes (like LF/HF), which appear to be devoid of intrinsic rate-dependency [16]. Thus, LF/HF ratio is considered the more appropriate to assess ANS independently of heart rate and was also the most powerful predictor of AF in our results. The intrinsic rate-independency of LF/HF is very important: indeed, in our study, heart rate at admission was also associated with new-onset AF on top of LF/HF.

### 4.3. AF Determinants in Acute Myocardial Infarction

The present findings also confirm the significance of several known clinical risk factors, such as age, CAD, blood pressure and heart rate for the development of AF. We found that, after adjustment, only age, HR on Holter recording, and LF/HF ratio remained significant predictors of AF. Indeed, our results show that patients with low a LF/HF ratio are three times more likely to develop AF than patients with a higher LF/HF ratio (OR 3.65 (2.20–0.76; *p* < 0.001)). The addition of LF/HF ratio to the classical AF predictors improved the diagnostic performance of the model, indicating that the parameters of the ANS are not fully covered by the classical clinical and biological predictors of AF during AMI. It therefore seems important to include this variable in existing risk scores. ANS plays an independent role on top of the variables classically used to predict AF, particularly on the parasympathetic side with a decrease in the LF/HF ratio. In addition, the other Holter ECG parameters used to evaluate parasympathetic activity, such as PNN50 or rMSSD, were consistent with our main hypothesis [9]. In our population, the AF group had higher PNN50 and rMSSD values, stressing the importance of the parasympathetic component as an AF substrate. This result has led us to speculate that an underlying chronic disease may have preceded the onset of AMI. Indeed, some authors have shown that parasympathetic tone is increased by expansion and atrial fibrosis in experimental models of AF [17]. In heart failure models, there is also an increase in sympathetic, parasympathetic and lymph node fibers in the left atrium that promote the maintenance and upkeep of arrhythmia [18]. These results are strong evidence that parasympathetic activation is related to the electrical remodeling of the LA through myocyte remodeling, and that parasympathetic activation occurs before the AMI [19]. Non-invasive measures obtained from Holter monitoring could identify an increased risk of AF in patients hospitalized for AMI, and could justify prolonged rhythm monitoring in patients identified as at-risk [20]. Moreover, in addition to AF prediction, HRV data on scopes could improve the stratification of ventricular rhythmic risk, as previously shown [21]. Consequently, we suggest that systematic and automated analysis of HRV data be added to the management of patients in the acute phase of MI. The contribution of LF/HF ratio to the prediction of AF recurrence risk at distance from the acute episode remains to be determined, but it may prove to be a useful tool for differentiating between acute AF episodes resulting from AMI and AF linked to pre-existing conditions at a high risk of recurrence [7].

### 4.4. Limitations

Among the 14,270 patients included in the RICO registry, 12,136 (85.0%) were excluded due to a lack of Holter monitoring. This could lead to a selection bias, considering that only patients for whom the hospital stay was scheduled to be >48h within our cardiology department were eligible to receive a Holter monitor, and this was also according to device availability. However, the large sample size of our included population, as well as the baseline characteristics that are close to the usual data obtained from the whole registry population in terms of age, CV risk factors and AMI type, should reduce this bias [22].

We were not able to provide the dimensions of LA for all patients, which is unfortunate seeing as these parameters are known to be associated with the onset of AF. In the subgroup of patients for whom LA parameters were available, we found that LA volume and LF/HF ratio were not collinear variables. This result suggests that the predictive value of LF/HF ratio on new-onset AF in the whole population is not likely to be related to a statistical relationship between the ANS and LA dimensions’ parameters.

Moreover, we can not exclude that new-onset AF patients had previous asymptomatic paroxysmal AF episodes (silent atrial fibrillation). However, we carefully checked patients medical history at admission to exclude previously known AF [3].

## 5. Conclusions

The results of our large HRV analysis indicate that autonomic dysregulation is strongly associated with new-onset AF in AMI. It would appear that AF in AMI is not related to an acute sympathetic activation, but rather on a parasympathetic one, suggesting the presence of chronically impaired cardiac autonomic regulation in patients who experience such events. However, in the absence of direct assessment of ANS activity, the causality between ANS dysfunction and increased risk of new-onset AF can only be inferred. Future studies are needed to test the clinical management of AMI patients guided by HRV [20].

## Figures and Tables

**Figure 1 jcm-09-01481-f001:**
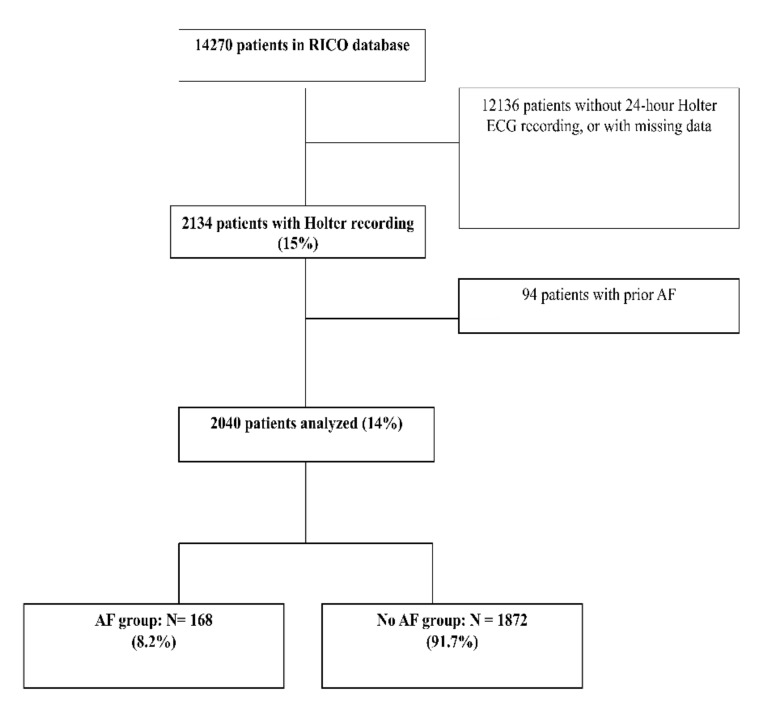
Study Flow Chart.

**Figure 2 jcm-09-01481-f002:**
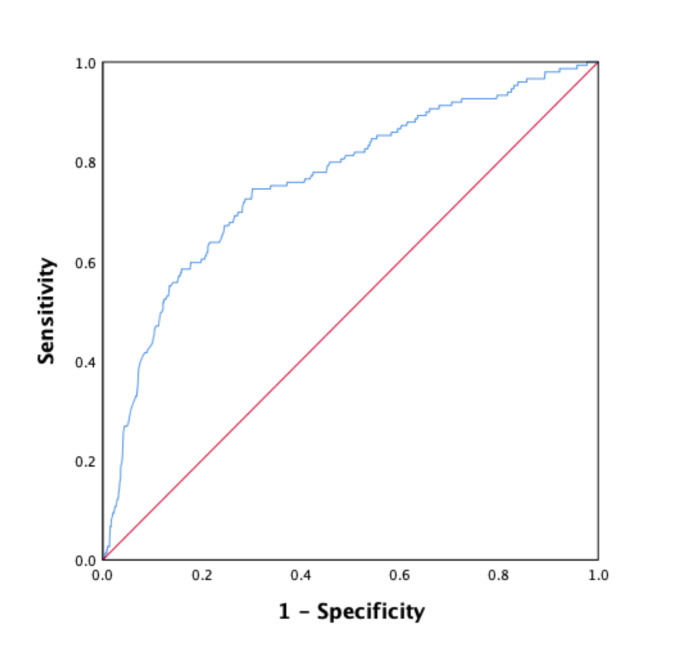
ROC curve demonstrating the predictive performance of LF/HF ratio for the onset of new AF during AMI: AUC = 0.73 (0.69–0.78; *p* < 0.001); optimal threshold: 1.735; sensitivity = 0.698; specificity = 0.707.

**Table 1 jcm-09-01481-t001:** Patient Baseline Characteristics (n (%) or median (interquartile range).

	No AF N = 1872	AF N = 168	*p*
Risk factors			
Age. years	64 (53–76)	77 (70–83)	<0.001
Female	519 (28%)	63 (38%)	0.007
BMI. kg/m^2^	26 (24–29)	26 (23–28)	0.067
Hypertension	924 (49%)	114 (68%)	<0.001
Hypercholesterolemia	832 (44%)	73 (44%)	0.804
Family history of CAD	503 (27%)	41 (24%)	0.489
Diabetes	388 (21%)	36 (21%)	0.830
Smoking	649 (35%)	25 (15%)	<0.001
CV history			
CAD	350 (19%)	40 (24%)	0.106
Stroke	99 (5%)	16 (10%)	0.023
Chronic renal failure	67 (4%)	8 (5%)	0.435
Clinical data on admission			
HR. beats/min	76 (65–89)	89 (70–108)	<0.001
SBP	140 (121–160)	130 (116–150)	<0.001
DBP	80 (70–92)	80 (64–90)	0.019
Heart failure (Killip >1)	315 (17%)	59 (35%)	<0.001
Anterior wall location	677 (36%)	58 (35%)	0.671
STEMI	1075 (57%)	107 (64%)	0.115
LVEF. %	55 (45–61)	47 (40–56)	<0.001
LVEF <40%	224 (13%)	33 (21%)	0.004
GRACE risk score	138 (115–163)	179 (150–202)	<0.001
Time to admission. min	192 (105–450)	180 (106–373)	0.818
ICU stay. days	4 (3–5)	5 (3–8)	<0.001
Biological data on admission			
Creatinine, μmol/l	87 (74–105)	98 (79–117)	<0.001
eGFR CKD, mL/min	76.3 (58.8–91.6)	60.9 (45.6–74.8)	<0.001
Glycaemia. mmol/L	7.00 (5.92–8.77)	7.92 (6.45–10.41)	<0.001
CRP ≥ 3 mg/L	1064 (63%)	111 (76%)	0.003
Troponin Ic peak, μg/L	18.5 (3.8–41.0)	20.5 (6.5–41.0)	0.204
NT–proBNP, pg/mL	542 (138–2177)	2450 (735–6915)	<0.001
Log NT–proBNP, pg/mL	2.73 (2.14–3.34)	3.39 (2.87–3.84)	<0.001
Chronic medication on admission			
Amiodarone	10 (1%)	5 (3%)	0.005
ARB/ACE inhibitors	633 (34%)	76 (45%)	0.002
Beta blockers	463 (25%)	63 (38%)	<0.001
Diuretic	408 (22%)	64 (38%)	<0.001
Antiplatelet	206 (11%)	25 (15%)	0.129
Aspirin	360 (19%)	51 (30%)	0.001
VKA	28 (2%)	5 (3%)	0.187
Statin	463 (25%)	46 (27%)	0.447
Acute medications <48 h			
Amiodarone	72 (4%)	58 (35%)	<0.001
ARB/ACE inhibitors	1413 (75%)	103 (61%)	<0.001
Beta blockers	1508 (81%)	109 (65%)	<0.001
Statin	1533 (82%)	121 (72%)	0.002

ACE: angiotensin conversion enzyme; ARB: angiotensin receptor blockers; AF: atrial fibrillation; BMI: body mass index; CAD: Coronary artery disease; CRP: C–reactive protein; CK: creatine kinase; CKD: chronic kidney disease; COPD: chronic obstructive pulmonary disease; DBP: diastolic blood pressure; eGFR: estimated glomerular filtration rate; HR: heart rate; ICU: Intensive Care Unit; LVEF: left ventricular ejection fraction; LWM: low molecular weight; NT-proBNP: N-terminal pro brain natriuretic peptide. PAD: peripheral artery disease; SBP: systolic blood pressure; STEMI: ST segment elevation myocardial infarction; VKA: vitamin K antagonist.

**Table 2 jcm-09-01481-t002:** Acute myocardial infarction management (n (%)).

	No AF N = 1872	AF N = 168	*p*
Invasive treatment			
Coronary angiography	1782 (95%)	154 (92%)	0.047
TIMI class on culprit artery	N = 1716	N = 150	0.308
0	789 (46%)	76 (51%)	
1	93 (5%)	7 (5%)	
2	179 (11%)	20 (13%)	
3	655 (38%)	47 (31%)	
TIMI class on culprit artery <2	882 (51%)	83 (55%)	0.355
CABG	71 (4%)	8 (5%)	0.533
Medical treatment			
Thrombolysis	274 (15%)	24 (14%)	0.902
Antiplatelet	1632 (87%)	135 (80%)	0.013
Aspirin	1811 (97%)	155 (92%)	0.003
Low molecular weight heparin	1238 (66%)	62 (37%)	<0.001
Unfractionated heparin	850 (45%)	116 (69%)	<0.001
Glycoprotein IIbIIIa inhibitors	798 (43%)	53 (32%)	0.005

AF: atrial fibrillation; CABG: coronary artery byass graft surgery; PCI: percutaneous coronary intervention.

**Table 3 jcm-09-01481-t003:** Holter parameters according to the onset of new AF during AMI, n (%), median (interquartile range) or mean (± standard deviation).

	No AF N = 1872	AF N = 168	*p*
Heart rate. beats/min	66 (60–73)	73 (60–84)	<0.001
Premature Ventricular Contractions (/24 h)	11 (2–92)	70 (7–378)	<0.001
VT episode (/24 h)	0 (0–0)/1 ± 22	0 (0–0)/4 ± 44	0.011
pNN50. %	4 (1–11)	11 (2–36)	<0.001
rMSSD. ms	27 (19–41)	45 (24–108)	<0.001
SDNN. ms	83 (64–107)	90 (61–119)	0.123
Power. ms^2^	1850 (925–3507)	2007 (1036–5171)	0.044
LF/HF	2.75 (1.46–4.58)	0.88 (0.57–2.00)	<0.001
LF/HF < 1.735	532 (30%)	111 (75%)	<0.001

AF: atrial fibrillation; VT: ventricular tachycardia; pNN50: proportion derived by dividing NN50 (the number of interval differences of successive NN intervals greater than 50 ms) by the total number of NN intervals; rMSSD: root mean square of successive differences in NN intervals; SDNN: standard deviation of all intervals between adjacent QRS complexes resulting from sinus node depolarization; LF: low frequencies; HF: high frequencies.

**Table 4 jcm-09-01481-t004:** In-hospital outcomes according to the onset of new AF during AMI n (%).

	No AF N = 1872	AF N = 168	*p*
Death	21 (1.1%)	8 (4.8%)	0.002
CV death	17 (0.9%)	6 (3.6%)	0.009
Recurrent MI	89 (4.8%)	10 (6.0%)	0.489
Heart Failure	417 (22%)	93 (55%)	<0.001
Stroke	19 (1.0%)	5 (3.0%)	0.042
VT or VF	130 (6.9%)	29 (17.3%)	<0.001

AF: atrial fibrillation; CV: cardio vascular; MI: myocardial infarction; VT: ventricular tachycardia; VF: ventricular fibrillation.

**Table 5 jcm-09-01481-t005:** Logistic regression analysis for the prediction of in-hospital AF.

	Univariate		Multivariable 1		Multivariable 2	
Characteristic	OR (95% CI)	*p*	OR (95% CI)	*p*	OR (95% CI)	*p*
Female	1.564 (1.126–2.172)	0.008	0.756 (0.501–1.140)	0.182	0.671 (0.434–1.038)	0.073
Age, Years	1.063 (1.049–1.077)	<0.001	1.049 (1.028–1.071)	<0.001	1.036 (1.01–1.060)	0.002
Hypertension	2.166 (1.547–3.032)	<0.001	1.160 (0.734–1.834)	0.525	1.234 (0.756–2.014)	0.400
Smoker	0.329 (0.213–0.509)	<0.001	0.996 (0.561–1.766)	0.988	1.078 (0.586–1.980)	0.810
Previous Stroke	1.885 (1.084–3.279)	0.025	0.994 (0.513–1.926)	0.986	0.949 (0.484–1.862)	0.879
HR (holter), bpm	1.043 (1.031–1.055)	<0.001	1.039 (1.025–1.054)	<0.001	1.034 (1.019–1.048)	<0.001
CRP >3 mg/L	1.785 (1.210–2.633)	0.003	1.189 (0.741–1.910)	0.473	1.199 (0.722–1.992)	0.482
eGFR CKD, mL/min	0.975 (0.969–0.981)	<0.001	0.997 (0.987–1.008)	0.616	0.998 (0.987–1.008)	0.659
NT–proBNP (log)	2.687 (2.143–3.369)	<0.001	1.479 (1.100–1.990)	0.010	1.379 (1.001–1.899)	0.049
Glycemia, mmol/L	1.000 (0.995–1.005)	0.998	X		X	
Troponin I Peak, µg/L	1.001 (0.997–1.004)	0.699	X		X	
Beta Blockers (Chronic)	1.826 (1.313–2.539)	<0.001	1.067 (0.701–1.624)	0.763	0.950 (0.604–1.493)	0.823
ARB/ACE Inhibitors (Chronic)	1.680 (1.222–2.311)	0.001	0.921 (0.601–1.411)	0.704	0.866 (0.550–1.361)	0.532
LF/HF <1.735	6.748 (4.605–9.889)	<0.001	X		3.377 (2.047–5.572)	<0.001
Quality Indexes		Phl = 0.324; −2LL = 796.132; %class = 92.5		pHL = 0.106;−2LL = 706.701; %class = 92.8		

ACE: angiotensin conversion enzyme; ARB: angiotensin receptor blockers; CRP: C-reactive protein; CKD: chronic kidney disease; eGFR: estimated glomerular filtration rate; HR: heart rate.

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
