# Peer review of "Involvement of Autonomic Nervous System in New-Onset Atrial Fibrillation during Acute Myocardial Infarction"

_jcm, 2020, doi:10.3390/jcm9051481_

Round 1
Reviewer 1 Report
In this study, Sagnard and colleagues found that new-onset atrial fibrillation among AMI patients was due to autonomic nervous system imbalance, as evident in low LH/HF ratio and higher PNN50 and RMSSD values by Holter ECG measurements. Some important shortcomings need to be addressed in this paper.
- Please avoid sentences such as „much lower“ etc., please state „significantly lower“ or more conventional scientific language. Please use this throughout the manuscript.
- How can you be certain that some of the patients with „new-onset AF“ post-MI did not have paroxysmal, non-documented episodes of AF before MI event?
- 16 (10%) of your patients in AF arm had a stroke at baseline. Could these strokes be due to AF? Do you have data for this? If this can be attributed to AF, then that means that such patients should be excluded from your new-onset AF group.
- Patients in the AF were older, more female, more hypertensive which is all well-established risk factors for the onset of AF.
- LF/HF <1.735 must be adjusted for additional covariates in the multivariable model such as NT-proBNP, beta-blocker dose and glycemia.
- Variables such as LVEF<40%, CRP >3 mg/L and eGFR <60 mL/min should be presented as continuous variables in the regression model rather than dichotomous variables, NT-proBNP and glycemia should be integrated as well.
- I think that a big problem in the regression analysis is those echo variables and baseline variables are separated when it comes to integrating the LF/HF ratio with those variables. I think that models from table 4 and elements discussed in the points 5,6 should be expanded for LA volume for example and this integrated regression model encompassing echo, laboratory and anthropometric variables should be reported with respective odds ratios
- It has been proved before that markers of myocardial stretch (NT-proBNP) and inflammation (CRP) were significantly associated with new-onset Afib after MI. Look at the paper by Parashar et al. 2013. Am J Cardiol. Why did you decide to discard NT-proBNP from your analysis? Especially given the fact that median NT-proBNP levels were nearly 5-fold higher in the new-onset AF group. Also goes for CRP à CRP likely higher in new-onset AF group compared to non-AF. Please be consistent in the table in the sense that both CRP, NT-proBNP and troponin values should be presented as continuous data (either as mean plus standard deviation or medians with IQR). Why did you choose to present some laboratory markers as dichotomous and some as continuous data in the first place? This is a huge inconsistency. Use all continuous variables as continuous variables because that is the most realistic and most honest approach to data rather than using arbitrarily set cut-offs and turning continuous data into nominal. Please fix this as this is a major limitation of this paper.
- What was the AMI treatment of these patients in terms of antiplatelet therapy, DAPT, etc.? Treatment at the time of AMI should be put into perspective and see if there are differences between the group?
- What was the TIMI flow rate between the groups and was it measured? Did you record a Killip class?
- You defined this cohort as an AMI cohort, meaning STEMI + NSTEMI. How were these patients treated for their AMI? This is unknown? What percentage was stented, underwent PCI, which proportion was perhaps treated more conservatively? These are all pertinent questions that are not answered nor presented in your Results section.
- The list of medications at baseline seems to arbitrarily. While about 50% of patients are hypertensive, no proportion of antihypertensive use (e.g. ACE inhibitor/ARBs, etc.) are discussed nor compared in the medications list of Table 1. This is all pertinent since it has an effect on LV remodeling and in some studies modified the trajectory of potential AF onset and sustain. Please include it and potentially adjust it for your analyses.
Author Response
Reviewer 1:
In this study, Sagnard and colleagues found that new-onset atrial fibrillation among AMI patients was due to autonomic nervous system imbalance, as evident in low LH/HF ratio and higher PNN50 and RMSSD values by Holter ECG measurements. Some important shortcomings need to be addressed in this paper.
We thank the reviewer 1 for the careful reading of the manuscript and for the revisions he/she asked that will certainly improve the manuscript quality
- Please avoid sentences such as „much lower“ etc., please state „significantly lower“ or more conventional scientific language. Please use this throughout the manuscript.
As suggested, we revised the manuscript to remove such sentences.
- How can you be certain that some of the patients with „new-onset AF“ post-MI did not have paroxysmal, non-documented episodes of AF before MI event?
The 2016 ESC guidelines state that “The diagnosis of AF requires rhythm documentation using an electrocardiogram (ECG) showing the typical pattern of AF: Absolutely irregular RR intervals and no discernible, distinct P waves” and that “Individuals with AF may be symptomatic or asymptomatic (‘silent AF’).”1
By definition, silent AF is undiagnosed because asymptomatic and not assessed using conventional ECG recordings2. Silent atrial fibrillation could indeed have occurred previously in our new-onset AF patients, and considering the association of new onset AF with LA remodeling parameters such as LA volume or ANS dysregulation, it is likely to have occurred. We added this limitation in the limitation chapter.
However, in order to classify the patients in the AF or sinus rhythm group, we checked for each patient admission his/her previous medical history, baseline treatments and medical recording in the hospital files. In case of suspicion of previous AF that cannot be confirmed by the patient (anticoagulant therapy, antiarrhythmic drugs) we systematically call the general practitioner or the cardiologist of the patient to explain this drug prescriptions. This quality check is since the beginning of our registry part of our routine procedure.
- 16 (10%) of your patients in AF arm had a stroke at baseline. Could these strokes be due to AF? Do you have data for this? If this can be attributed to AF, then that means that such patients should be excluded from your new-onset AF group.
The reviewer raises a very important question, that we partly answered in the previous question. Indeed, to be sure that we excluded previous AF, we had a very meticulous quality check process detail in the question 2. We can therefore be confident that no previously known AF patient was included in our study. We added the following sentence in the methods chapter : we checked for each patient admission his/her previous medical history, baseline treatments and medical recording in the hospital files. In case of suspicion of previous AF that cannot be confirmed by the patient (anticoagulant therapy, antiarrhythmic drug) we systematically called the general practitioner or the cardiologist of the patient to explain this drug prescriptions..
- Patients in the AF were older, more female, more hypertensive which is all well-established risk factors for the onset of AF.
Indeed, the AF group depicted previous history that suggests an atrial substrate prone to trigger AF, as highlighted also by ANS dysregulation and larger LA volumes. These results strengthen the hypothesis that “acute AF” is not an epiphenomenon due to the acute event but rather an opportunity to screen for previously silent AF that is unmasked by the AMI. In a previous work we proved that new-onset AF (silent as well as symptomatic) during AMI is likely to reoccur at follow-up 3.
- LF/HF <1.735 must be adjusted for additional covariates in the multivariable model such as NT-proBNP, beta-blocker dose and glycemia.
AND
- Variables such as LVEF<40%, CRP >3 mg/L and eGFR <60 mL/min should be presented as continuous variables in the regression model rather than dichotomous variables, NT-proBNP and glycemia should be integrated as well.
In accordance to the reviewer’s suggestion, we performed a new multivariate analysis including NTproBNP, betablockers and ACE inhibitors /ARB previous treatment, glycemia. Moreover we used eGFR and LVEF as continuous variables as requested. However, as regards to CRP, it is not possible to use it as a continuous variable since the normal value is just displayed as “<3” in our center. The distribution of such variable then is not continuous.
Moreover, as NTproBNP and LVEF are collinear variables, we did not put them together in the multivariable analysis but instead we built 2 multivariable models. In both models, LF/HF ratio<1.735 had an incremental value when added to the multivariable model (p<0.001) and the independent predictors of new onset AF were age, mean heart rate on Holter, and LF/HF<1.735. Moreover, in the model including log NT-proBNP, this variable was also an independent predictor of AF.
LVEF was not an independent predictor of AF when added into the multivariable model instead of NT-proBNP (OR 0.997 (0.981-1.013), p=0.730).
Interestingly, the predictive value of LF/HF ratio was independent of all classical predictors of AF included in the model, even after multiple adjustments.
The new multivariable models have been added in the results instead of the previous one and the results have been changed accordingly.
- I think that a big problem in the regression analysis is those echo variables and baseline variables are separated when it comes to integrating the LF/HF ratio with those variables. I think that models from table 4 and elements discussed in the points 5,6 should be expanded for LA volume for example and this integrated regression model encompassing echo, laboratory and anthropometric variables should be reported with respective odds ratios
We acknowledge that this subgroup analysis was confusing and lacks of precision. In fact, this analysis was designed to test if the predictive value of LF/HF ratio on new-onset AF in the whole population could be related to a statistical relationship between ANS and LA dimensions parameters. We did not had the statistical power to include more than 2 variables in the multivariate analysis to adjust for classical predictors of AF unfortunately.
In accordance to the reviewer’s remark, we thus removed the table 5, and we added the following sentences :
In the results chapter: Among the patients included in the subgroup analysis, 8 had a new-onset of AF during AMI. However, after bivariate analysis, neither LA volume nor LH/HF remained significantly associated with AF (p=0.062 for both variables). Collinearity between the variables was not significant (Variation Inflation Factor =1.07).
In the limitation chapter : We were not able to provide the dimensions of LA for all patients, which is unfortunate seeing as these parameters are known to be associated with the onset of AF In the subgroup of patients for whom LA parameters were available, we found that LA volume and LF/HF ratio were not collinear variables. This result suggests that the predictive value of LF/HF ratio on new-onset AF in the whole population is not likely to be related to a statistical relationship between ANS and LA dimensions parameters.
- It has been proved before that markers of myocardial stretch (NT-proBNP) and inflammation (CRP) were significantly associated with new-onset Afib after MI. Look at the paper by Parashar et al. 2013. Am J Cardiol. Why did you decide to discard NT-proBNP from your analysis? Especially given the fact that median NT-proBNP levels were nearly 5-fold higher in the new-onset AF group. Also goes for CRP à CRP likely higher in new-onset AF group compared to non-AF. Please be consistent in the table in the sense that both CRP, NT-proBNP and troponin values should be presented as continuous data (either as mean plus standard deviation or medians with IQR). Why did you choose to present some laboratory markers as dichotomous and some as continuous data in the first place? This is a huge inconsistency. Use all continuous variables as continuous variables because that is the most realistic and most honest approach to data rather than using arbitrarily set cut-offs and turning continuous data into nominal. Please fix this as this is a major limitation of this paper.
We are sorry to read that the reviewer thinks that the use of dichotomous variables instead of continuous ones could be scientifically doubtful. We chose to dichotomize these variables according to literature-based known cut-offs or ROC curves as we classically perform in our registry analyses, in order to increase the clinical relevance and the power of the statistical analysis. However, we removed all dichotomous variables as requested, with the exception of CRP as explained in the answer to your questions 5-6. The independent predictive value of LF/HF ratio remained unaltered after these multiple adjustments thus proving the strong interest of this ratio measurement to understand the pathophysiology of new onset AF during AMI.
- What was the AMI treatment of these patients in terms of antiplatelet therapy, DAPT, etc.? Treatment at the time of AMI should be put into perspective and see if there are differences between the group?
At a first time, we had to reduce the number of variables to match journal requirements in terms of format. However, we agree that this data is important to put our results into perspective of the AMI severity. We added this analysis to table 1.
- What was the TIMI flow rate between the groups and was it measured? Did you record a Killip class?
The TIMI flow in the culprit artery has been added in the new table 2 on angiographic data
In the table 1, heart failure at admission was defined by Killip >1. We added this clarification in the table.
- You defined this cohort as an AMI cohort, meaning STEMI + NSTEMI. How were these patients treated for their AMI? This is unknown? What percentage was stented, underwent PCI, which proportion was perhaps treated more conservatively? These are all pertinent questions that are not answered nor presented in your Results section.
We thank the reviewer for this relevant comment and we have built a new table (Table 2) showing the AMI management of patients in the AF and non-AF group.
- The list of medications at baseline seems to arbitrarily. While about 50% of patients are hypertensive, no proportion of antihypertensive use (e.g. ACE inhibitor/ARBs, etc.) are discussed nor compared in the medications list of Table 1. This is all pertinent since it has an effect on LV remodeling and in some studies modified the trajectory of potential AF onset and sustain. Please include it and potentially adjust it for your analyses.
As explained in our answer the question 9, a first time, we had to reduce the number of variables to match journal requirements in terms of format. However, we fully agree that this data is important so we added these results to table 1. Moreover, as answered to the questions 5-6, we adjusted our analysis on these variables and LF/HF ratio remained an important predictor of AF.
References :
- Kirchhof P, Benussi S, Kotecha D, et al. 2016 ESC Guidelines for the management of atrial fibrillation developed in collaboration with EACTS: The Task Force for the management of atrial fibrillation of the European Society of Cardiology (ESC)Developed with the special contribution of the European Heart Rhythm Association (EHRA) of the ESCEndorsed by the European Stroke Organisation (ESO). Europace. 2016.
- Guenancia C, Garnier F, Fichot M, Sagnard A, Laurent G, Lorgis L. Silent atrial fibrillation: clinical management and perspectives. Future Cardiol. 2020;16:133-142.
- Guenancia C, Toucas C, Fauchier L, et al. High rate of recurrence at long-term follow-up after new-onset atrial fibrillation during acute myocardial infarction. Europace. 2018;20:e179-e188.
Reviewer 2 Report
The known predictors of atrial fibrillation develop during the hospital period with acute myocardial infarction include older age, particularly more than 70 years , followed by increased of Body Mass Index , enlarged diameter of left atrium (LA), presentation of mitral regurgitation and B-type natriuretic peptide. This predictors may also strongly associated with dysregulated autonomic nervous system. In this article, AF patients were older, had more frequent hypertension and lower LVEF. Logistic regression analysis also include this confounding factors include age, LA diameter, hypertension and LVEF< 40%. Body Mass Index and B-type natriuretic peptide were not included in these analysis. My suggestion is that authors should consider use propensity score matching these known predictors for AF and non-AF patients to estimate the true effect of predicting in-hospital AF of autonomic dysregulation.
Author Response
Reviewer 2:
The known predictors of atrial fibrillation develop during the hospital period with acute myocardial infarction include older age, particularly more than 70 years , followed by increased of Body Mass Index , enlarged diameter of left atrium (LA), presentation of mitral regurgitation and B-type natriuretic peptide. This predictors may also strongly associated with dysregulated autonomic nervous system. In this article, AF patients were older, had more frequent hypertension and lower LVEF. Logistic regression analysis also include this confounding factors include age, LA diameter, hypertension and LVEF< 40%. Body Mass Index and B-type natriuretic peptide were not included in these analysis.
We thank the reviewer 2 for the careful reading of the manuscript and for the revisions he/she asked that will certainly improve the manuscript quality
- My suggestion is that authors should consider use propensity score matching these known predictors for AF and non-AF patients to estimate the true effect of predicting in-hospital AF of autonomic dysregulation.
In accordance tot the reviewer’s interesting remark, we performed an additional analysis using propensity score matching that confirms our results on the role of LH/HF ration in AF prediction. We added these sentences in the manuscript:
Methods: To improve the robustness of results, the AF patients were matched 1:1 with the SR patients using nearest neighbor matching on the linear propensity score with a tolerance of 0.02. The same logistic regression model was used to compare AF with SR patients.
Results: Moreover, after 1:1 propensity score matching (on age, sex, previous hypertension, previous stroke, BMI, LVEF), LF/HF ratio<1.735 (odds ratio [OR]: 3.49, 95% confidence interval [CI]: 2.03-5.99, p<0.001) remained independently associated with the
Reviewer 3 Report
To the authors:
I enjoyed reviewing your manuscript, entitled “Involvement of autonomic nervous system in new-onset atrial fibrillation during acute myocardial infarction.” This single-center retrospective cohort study of 2035 patients with acute myocardial infarction (AMI) compared heart rate variability in the 168 patients who developed new-onset atrial fibrillation (AF) with those who remained in sinus rhythm. AF patients had higher pNN50 values (number of intervals differences of successive NN intervals >50 ms / total number of NN intervals) and a lower median LH/HF ratio. In multivariate analysis, low frequency / high frequency RR interval ratio ratio (LF/HF) <1.735, increasing age, and mean sinus rhythm heart rate predicted AF. The manuscript concludes that AF in AMI occurs in the setting of autonomic dysregulation.
While prior studies have identified increasing age, male gender, diabetes, and congestive heart failure as risk factors for AMI-associated new-onset AF, this study is truly novel in its assessment of ECG-based autonomic nervous system parameters to uncover the etiology and risk factors for this common arrhythmia.
The manuscript could be improved with attention to the following details:
- Experimental Section, 2.1 Patients and 2.5 Holter ECG data. “The present study included all consecutive patients…who underwent a 24-hour Holter ECG recording during their coronary care unit stay.” In Figure 1, it appears that 14,270 patients with AMI were admitted and 12,136 (85.0%) were excluded due to a lack of EKG monitoring. In the Experimental Section, the authors should explain how AMI patients were selected for this monitoring, and in the Discussion, they should address the potential selection bias introduced by excluding a vast majority of the potential study population.
- Results, 3.1 Patient characteristics. “Of the 2134 patients in the RICO database, 2040 met the inclusion criteria.” As per above, this statement is somewhat misleading: actually, of 14,270 patients, 2040 (14.3%) met the inclusion criteria of having a 24-hour Holter ECG recording and no prior AF. This statement should be revised.
- Results, Tables 1 and 2. Because many readers focus on the tables and figures and do not read every single word of the text, it is important that the tables and figures be self-explanatory. In table 1, the numbers alternately represent medians with interquartile ranges, numbers of patients with percentages, and means with standard deviations. The figure headings or caption should explain which of these parameters the numbers represent.
- Results, Figure 2. The captions reports “Sensibility;” this should read “Sensitivity.”
- Results, 3.5 Echocardiographic parameters of left atrium. “In univariate analysis, the only LA size parameter that was a predictor of in-hospital AF was LA volume.” This statement seem somewhat misleading because, possibly given the small sample size, both LA volume and LF/HF<1.735 were not significant predictors on bivariate analysis, which is not mentioned in the text. The authors should state in the text that neither of these factors remained a statistically significant predictor on bivariate analysis.
Author Response
Reviewer 3:
I enjoyed reviewing your manuscript, entitled “Involvement of autonomic nervous system in new-onset atrial fibrillation during acute myocardial infarction.” This single-center retrospective cohort study of 2035 patients with acute myocardial infarction (AMI) compared heart rate variability in the 168 patients who developed new-onset atrial fibrillation (AF) with those who remained in sinus rhythm. AF patients had higher pNN50 values (number of intervals differences of successive NN intervals >50 ms / total number of NN intervals) and a lower median LH/HF ratio. In multivariate analysis, low frequency / high frequency RR interval ratio ratio (LF/HF) <1.735, increasing age, and mean sinus rhythm heart rate predicted AF. The manuscript concludes that AF in AMI occurs in the setting of autonomic dysregulation.
While prior studies have identified increasing age, male gender, diabetes, and congestive heart failure as risk factors for AMI-associated new-onset AF, this study is truly novel in its assessment of ECG-based autonomic nervous system parameters to uncover the etiology and risk factors for this common arrhythmia.
We thank the reviewer 3 for the careful reading of the manuscript and for the revisions he/she asked that will certainly improve the manuscript quality
The manuscript could be improved with attention to the following details:
- Experimental Section, 2.1 Patients and 2.5 Holter ECG data. “The present study included all consecutive patients…who underwent a 24-hour Holter ECG recording during their coronary care unit stay.” In Figure 1, it appears that 14,270 patients with AMI were admitted and 12,136 (85.0%) were excluded due to a lack of EKG monitoring. In the Experimental Section, the authors should explain how AMI patients were selected for this monitoring, and in the Discussion, they should address the potential selection bias introduced by excluding a vast majority of the potential study population.
We acknowledge that our formulation was inaccurate : we added the following sentence into the experimental section:
Holter monitoring was based on device availability, patient consent and expected hospital stay>48 hours within our cardiology department to be able to obtain Holter results before hospital discharge.
Moreover, in the limitation chapter, we added the following sentence: Among the 14,270 patients included in the RICO registry, 12,136 (85.0%) were excluded due to a lack of Holter monitoring. This could lead to a selection bias, considering that only patients in whom the hospital stay was scheduled to be >48h within our cardiology department were eligible to receive a Holter monitor, according to device availability also. However, the large sample size of our included population, as well as the baseline characteristics that are close to the usual data obtained from the whole registry population in terms of age, CV risk factors and AMI type, should reduce this bias1.
- Results, 3.1 Patient characteristics. “Of the 2134 patients in the RICO database, 2040 met the inclusion criteria.” As per above, this statement is somewhat misleading: actually, of 14,270 patients, 2040 (14.3%) met the inclusion criteria of having a 24-hour Holter ECG recording and no prior AF. This statement should be revised.
We apologize for these inaccuracies and we change the sentence as follows : “Of the 14,270 patients in the RICO database, 2,040 (14.3%) met the main inclusion criteria, of having a 24-hour Holter ECG recording and no prior AF”.
- Results, Tables 1 and 2. Because many readers focus on the tables and figures and do not read every single word of the text, it is important that the tables and figures be self-explanatory. In table 1, the numbers alternately represent medians with interquartile ranges, numbers of patients with percentages, and means with standard deviations. The figure headings or caption should explain which of these parameters the numbers represent.
We apologize for this missing data and have added the legends as requested.
- Results, Figure 2. The captions reports “Sensibility;” this should read “Sensitivity.”
We thank the reviewer for this comment, the change has been performed
- Results, 3.5 Echocardiographic parameters of left atrium. “In univariate analysis, the only LA size parameter that was a predictor of in-hospital AF was LA volume.” This statement seem somewhat misleading because, possibly given the small sample size, both LA volume and LF/HF<1.735 were not significant predictors on bivariate analysis, which is not mentioned in the text. The authors should state in the text that neither of these factors remained a statistically significant predictor on bivariate analysis.
We acknowledge that this analysis was confusing and lacks of precision. In fact, this subgroup analysis was designed to test if the predictive value of LF/HF ratio on new-onset AF in the whole population could be related to a statistical relationship between ANS and LA dimensions parameters.
In accordance to the reviewer’s remark, we removed the former table 5, and we added the following sentences :
In the results chapter: Among the patients included in the subgroup analysis, 8 had a new-onset of AF during AMI. However, after bivariate analysis, neither LA volume nor LH/HF remained significantly associated with AF (p=0.062 for both variables). Collinearity between the variables was not significant (Variation Inflation Factor =1.07).
In the limitation chapter : We were not able to provide the dimensions of LA for all patients, which is unfortunate seeing as these parameters are known to be associated with the onset of AF In the subgroup of patients for whom LA parameters were available, we found that LA volume and LF/HF ratio were not collinear variables. This result suggests that the predictive value of LF/HF ratio on new-onset AF in the whole population is not likely to be related to a statistical relationship between ANS and LA dimensions parameters.
References :
- Farnier M, Salignon-Vernay C, Yao H, et al. Prevalence, risk factor burden, and severity of coronary artery disease in patients with heterozygous familial hypercholesterolemia hospitalized for an acute myocardial infarction: Data from the French RICO survey. J. Clin. Lipidol. 2019;13:601-607.
Round 2
Reviewer 1 Report
I would wish to congratulate the authors on answering all my concerns effectively.